# Impact of Artificial Intelligence News Source Credibility Identification System on Effectiveness of Media Literacy Education

**Tosti H. C. Chiang** *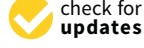**, Chih-Shan Liao and Wei-Ching Wang**

Graduate Institute of Mass Communication, National Taiwan Normal University, Taipei 106, Taiwan; csliao@ntnu.edu.tw (C.-S.L.); weiching@ntnu.edu.tw (W.-C.W.)
* Correspondence: tosti.chiang@gmail.com

**Abstract:** During presidential elections and showbusiness or social news events, society has begun to address the risk of fake news. The Sustainable Development Goals 4 for Global Education Agenda aims to "ensure inclusive and equitable quality education and promote lifelong learning opportunities for all" by 2030. As a result, various nations have deemed media literacy education a required competence in order for audiences to maintain a discerning attitude and to verify messages rather than automatically believing them. This study developed a highly efficient message discrimination method using new technology using artificial intelligence and big data information processing containing general news and content farm message data on approximately 938,000 articles. Deep neural network technology was used to create a news source credibility identification system. Media literacy was the core of the experimental course design. Two groups of participants used different methods to perform message discrimination. The results revealed that the system significantly expanded the participants' knowledge of media literacy. The system positively affected the participants' attitude, confidence, and motivation towards media literacy learning. This research provides a method of identifying fake news in order to ensure that audiences are not affected by fake messages, thereby helping to maintain a democratic society.

**Keywords:** artificial intelligence; media literacy education; news source credibility identification; learning effectiveness; learning attitude

## 1. Introduction

*Research Background*

The Sustainable Development Goals (SDG) 4 for Global Education Agenda aims to "ensure inclusive and equitable quality education and promote lifelong learning opportunities for all" by 2030. The sharing mechanisms of social platforms allow for users to share messages rapidly and in short intervals. The public can integrate the content from different platforms on the basis of the nature of the content [1]. Thus, similar situations can be observed in content farm messages. To create a larger-scale reposting effect, messages often have sensational headlines to increase the reader click rate.

Content farms are a form of web page that produces large quantities of low-quality articles on a variety of topics. They use keywords to increase the ranking of the result pages in various search engines. The business model of content farms is based on the placement of commercials or the sale of certain items on the pages or in the content to generate revenue. In terms of the benefit to business, a webpage's profit derives from the quantity of the internet flow; the higher the flow, the more the profit. To attract more readers to view pages, content farms create articles on highly popular topics and add exaggerated titles and content, such as modified or fake images. As a result, readers read articles without knowing the truth [2].

Content farms usually produce articles that (1) may be short and not in the format of news reports, with simple language and few citations or links; (2) are filled with advertisements; (3) contain content from other websites that has been copied or modified; and (4) contain many external links [3]. These features indicate that the content of the website is not credible. However, general audiences have difficulty determining credibility because of the quantity of the information, and often accept the information as true.

Messages that have been plagiarized and reposted without authorisation lose their credibility. In addition to being false, the articles can have a substantial effect if their content is used to advocate for a specific purpose, event, or person. During times of disaster, this can cause social unrest [4]. In addition, content farms can be hired to make politicians and their teams appear more popular. With the growing influence of social media, the manipulation of data through social media has become an efficient method to influence users [5].

Because of the nature of content farm messages, evaluating messages is a complex and multifaceted task. Strategies can be used to determine whether messages are credible in terms of their content, writing style, dissemination path, and organisational credibility. News-related features (such as the title, contents, and author of an article) and social features (such as response, dissemination path, and platform) can be used perform message discrimination. However, messages can comprise text, multimedia, or internet articles, and therefore require appropriate techniques and resources [6].

Educational organisations in different fields have used media literacy to establish a set of message discrimination directions. Audiences can use the fact-checking steps provided by the Harvard University Library [7] and the International Federation of Library Associations (IFLA) [8] to maintain a discerning attitude towards messages and perform a multifaceted verification their sources with available tools. However, the above-mentioned method indicates that discriminating among messages is a complex process involving the use of corresponding techniques to discern the type of message. Message discrimination is not an easy task. The method assesses the audience's media literacy and determines whether they can maintain a discerning attitude and perform message discrimination.

Because of the difficulty of discriminating among messages, various countries have invested in research on artificial intelligence (AI) learning and discrimination to help audiences. In addition, the contents and titles of content farm messages have identifiable features that distinguish them from truthful messages [9]. In 2019, the University of Washington developed the Grover message discrimination system, which uses deep learning to learn the features of fake messages. This system has a message discrimination accuracy of 92% [10].

However, no research has been conducted on the use of AI systems for the discrimination of Chinese messages. In addition, although an English-language message database has been created, no similar database has been created for Chinese content. Thus, this study used a system based on the collection of data from the internet (comprising Uniform Resources Locators (URLs), domains, and web address information) and established a terms database. The content of news articles and news-related information was disassembled and analysed to construct a model with AI learning to create a message discrimination system for Chinese content.

The AI news source credibility identification system developed in this study discriminates among messages on the basis of writing style. In terms of writing style, content farm messages have a higher proportion of adjectives and adverbs than regular news. These features can be used as a basis for helping the audience to discriminate information and for helping the system to learn.

In addition to developing the system, this study conducted a media literacy course. The utilisation of the system was the basis for determining whether the use of the system for learning media literacy positively affected the users' learning effectiveness and attitudes toward media literacy. The researchers designed both the courses and the test items to evaluate the users' results. The questionnaires were based on learning attitude theory to

determine the users' attitude towards learning media literacy after they used the system as a discrimination tool.

Based on the aforementioned abstract introduction and this study's objective, this study proposed the following two research questions:

1. After using the AI news source reliability identification system, did the experimental group have higher learning effectiveness than the control group?
2. After using the AI news source reliability identification system, did the experimental group have a more positive learning attitude than the control group?

## 2. Literature Review

### 2.1. Content Farm Messages and Their Impact

According to Google's Search Quality Raters Guidelines [11], search engines define content farm websites as low-quality sites. Low-quality sites manipulate search engine rankings to achieve high-volume message reposting, and have three main features: (1) automatically generated content; (2) a high volume of outgoing and incoming link manipulation; and (3) little or no original content. The objective of such sites is to manipulate search engine ranking results in order to achieve high-volume message reposting.

Writers hired by content farms are usually not professional reporters. Because of operating costs, content farms often hire freelancers without professional news writing backgrounds. Horne and Adalı (2017) compared large quantities of data to identify the following features of fake news articles: (1) longer titles than those of real news articles, which attract attention; (2) less content and more redundancy, adjectives, and adverbs than real news articles; and (3) more colloquial language, along with the use of 'you' to address readers and induce self-suggestion.

Content farms quickly produce large quantities of articles that are often plagiarised and pieced together from multiple sources and that contain half-truths and popular keywords. Therefore, false messages are often disseminated in large quantities after major social events, creating social turmoil. The operational strategy of content farms is to hire low-paid writers, produce a large number of articles, and utilise search engine optimisation (SEO) technology to increase advertisement revenue [4].

However, content farms do not only want to derive revenue from advertising. They disseminate information to gain popularity and influence on the internet. In Taiwan, public attention to fake news has increased. Fake news is perceived as having a strong effect that has caused an information war. Messages from Chinese content farms can affect Taiwan and the perception of Taiwanese people in the rest of the world. Long-term exposure to half-truths and inaccurate messages can negatively affect citizens' critical thinking and ability to reflect, thereby causing homophily. Homophilic audiences tend to believe in certain content more easily and reaffirm what they already believe, which decreases the opportunity for social discussions of topics such as politics and engenders feelings of animosity [12].

### 2.2. Media Literacy Manual Discrimination Method

Media literacy has been discussed quite a lot in recent years, but the concept of media literacy was actually put forward in a book published by Christ College, Cambridge University, UK in 1950 [13]. At that time, traditional mass media such as radio and television were in a period of vigorous development and the United Kingdom began to incorporate media into the school curriculum [14]. Hobbs R. [15] mainly focused on training K-12 students to understand questions such as "Who produces media contents?", "What is the form of media texts?", "How are media texts produced?", "How do readers understand contents?", "Who are the readers?", "How does the content reproduce the truth?" and more. In recent years, due to the rapid development of digital technology, media literacy has evolved into digital literacy. Digital literacy education includes "understanding the content and symbolic characteristics of media messages, speculating on media representation,

reflecting on the meaning of readers, analyzing media organization, influence and recent Using Media" [16].

Media literacy is the ability to help the public resist the dangers of false information, however, it is not easy to cultivate media literacy education. Although European and American countries began to attach importance to media literacy education very early, countries such as the United Kingdom, Canada, Australia, and other countries have added media education to their formal education system as well; however, in the face of today's fake news problem the effect is limited.

During the discrimination process, audiences discuss content within themselves and voice doubts regarding content. While this message-receiving behaviour may differ from past behaviours regarding receiving information, doubt remains a prerequisite to the ability to determine fake news and allows audiences to analyse and criticise information rather than automatically subscribing to the ideology behind the message [17].

Therefore, media literacy education must be used to cultivate discrimination skills. According to the Harvard Library [7], message inspection consists of five steps: (1) reviewing sources; (2) verifying URLs and websites; (3) evaluating the appearance of fake news websites, which tend to be poorly designed; (4) referring to other recommended resources if the content arouses anger or confusion; and (5) installing browser plug-ins to block known fake news.

The IFLA (2016) provides eight steps to discriminate fake news: (1) considering the news source; (2) checking the author; (3) checking the date; (4) self-reflecting; (5) understanding the intent behind the message; (6) referencing other data; (7) identifying satirical articles; and (8) asking experts. These steps can be used at the moment of receiving news.

### 2.3. AI Credibility Identification System

To address the abundance of information, this study constructed a model that enabled machines to learn to identify messages using several criteria. The system consists of a human part and a machine part. The human part is related to methods provided by fact-checking organisations, and the machine part is related to a language database and news database.

The main purpose of creating a news database was to categorise news using feature labelling. This method is used to determine whether news is real or fake as well as to identify the writing style and other features of an article in order to discriminate its contents. The most efficient identification model is a human–machine hybrid discrimination method combining criteria fit for human judgement and data from language and news databases. Humans and machines each have certain disadvantages. Humans cannot process a large amount of data at once, and machines cannot understand the meaning of words without a learning process. Our AI-based fake news discrimination method combines the advantages of humans and machines into a human–machine hybrid model. Humans combine their media literacy and understanding of words with the system computations to perform a mining analysis of language and the internet [18].

Depending on the discriminating features and technology, fake news discrimination methods generally comprise human and machine labour. The methods are categorised into the following four parts: knowledge-based analysis, in which errors in the content of fake news are analysed; style-based analysis, in which the writing style of fake news is analysed; propagation-based analysis, in which the propagation model of fake news is analysed; and credibility-based analysis, in which the credibility of the source, spreaders, and propagation organisations of fake news is analysed [19]. This study created a human–machine hybrid operational environment using knowledge-based analysis, style-based analysis, and human and machine labour.

The knowledge-based analysis was divided into human and automatic monitoring. Human monitoring refers to centres that fact-check through human labour. Automatic monitoring requires the development of information retrieval (IR) and natural language processing (NLP) technology. Automatic monitoring processes large-scale data more

effectively. Automatic monitoring requires IR and fact-checking stages. The purpose of the IR stage is to establish a knowledge database. Words (subject–predicate–object) are classified on the basis of the data before the fact-checks are performed. The most common method of discriminating among writing styles is to use features such as the analytic dimensions. This method requires professional journalism knowledge. Writing styles differ between general news and content farm messages. Fake news outbreaks follow high-profile events (e.g., natural disasters). Using human labour to identify these messages immediately following these events in large quantities has limitations. Therefore, machines must be trained to perform real-time analysis and monitoring through deep learning.

*2.4. Computer-Aided Instruction and Learning Attitude Assessment*

This study utilized problem-solving computer-aided learning, which guides learners to find solutions to their problems. In daily life, problem-solving is an open-ended process in which learners must analyse a problem and develop a hypothesis to solve the problem. In problem solving-oriented learning, instructors use real life examples to encourage discussion among students and strengthen their abilities to think, discuss, and criticise in order to solve problems. Ideally, such courses increase students' learning motivation and integrate long-term knowledge databases [20].

Studies have indicated that computer-aided learning increases the effectiveness of language learning. In modern education, computer-aided learning has caused the conventional model of English audiovisual teaching to shift towards a multimedia teaching model. By creating a motivational learning environment, students can advance from learning English to participating in the teaching of English. The teacher then becomes a promoter of learning, organiser of classroom activities, designer, and inquirer, monitoring and assessing messages and resources [21].

"Attitude" is an individual's assessment of a target event or behaviour and their positive or negative emotions while pursuing a goal. Attitude represents an individual's evaluation of an event, object, or other individual. Therefore, attitude can be measured and quantified in order to predict students' learning performance and achievements while taking into account individual differences [22].

Several studies have measured attitude in students learning various subjects. To measure attitude in science education, Pulungan and Nasution [23] adopted two different types of teaching methods. The experimental group received a scientific inquiry learning model, whereas the control group received the regular learning method. The study indicated that the students who received the scientific inquiry learning model were more successful than were those who received the regular learning methods. In addition, the students with a more positive attitude towards science outperformed those with negative attitudes towards science. Therefore, student attitudes towards learning are positively correlated with learning outcomes.

To quantify attitude as a variable, studies on natural science teaching have divided attitude into two categories, namely, scientific attitudes and attitudes towards science [24]. Attitude towards science refers to students' attitude towards a subject as well as their mental state, interest, motivation, and anxiety. This study applied the concept of attitude towards science to media literacy in order to determine students' attitude towards and interest in media literacy after the course.

**3. Research Methodology**

After integrating various discussions in research papers and relevant theories, this study administered a comprehensive test to assess the effectiveness of users' hands-on experience with the AI news credibility identification system.

### 3.1. AI Fake News Identification System

### 3.1.1. Collection of AI News Credibility Data

To build the database used in this study, we used content farms as the main source for fake news data because content farms are the primary source of fake news. For general news, this study used the four news outlets most trusted by the Taiwanese public: Apply Daily, China Times, United Daily News, and Liberty Times, along with the electronic version of the Central News Agency (CNA), to ensure the quality of news. These four news outlets were selected as the sources of general news because of their credibility ranking in the "2019 Taiwan News Media Credibility Research" published by Taiwan Media Watch. The electronic versions of these four major news media sources are generally accepted by the Taiwanese public. As of March 2021, the database comprised approximately 850,000 general news articles (Table 1).

**Table 1.** Normal news in dataset.

| Media | Number of Data | Collection Time |
|---|---|---|
| Apple daily | 91,194 | Sep. 2019–Mar. 2021 |
| China times | 127,922 | Sep. 2019–Mar. 2021 |
| United daily news | 585,584 | Sep. 2019–Mar. 2021 |
| Liberty times | 51,874 | Sep. 2019–Mar. 2021 |

Fake news collection was collected from five well-known content farms, namely, The Global Times, Mission, Nooho, Kknews, and Qiqi.news. These sites were selected because their content, particular that of Mission and Nooho, is widely reproduced on Facebook. In addition, kknews consistently has a high search engine ranking. As of March 2021, the content database comprised approximately 88,000 data points (Table 2).

**Table 2.** News from content farms in dataset.

| Media | Number of Data | Collection Time |
|---|---|---|
| The Global Times | 6036 | Dec. 2017–Mar. 2021 |
| Mission | 9425 | June 2019–Mar. 2021 |
| Nooho | 1749 | July 2019–Mar. 2021 |
| Kknews | 40,860 | Oct. 2019–Mar.2021 |
| Qiqi.news | 30,252 | Nov. 2019–Mar.2021 |

The fields used in the database were (1) title name; (2) news code; (3) source; (4) date; (5) title; (6) content; (7) URL; and (8) label. The title and content columns were required fields in the system to create a standard for discrimination. Studies have revealed that the titles of fake news articles are generally longer than those of ordinary news articles because the creators of fake news articles attempt to summarise the contents in the title to increase the speed of message relay and change audiences' message reception behaviours [7]. Hence, the title and content were required fields in the system, whereas author, source, and date of publication were not.

In addition to Taiwan Media Watch reports on the general news outlets, reputation in news media and whether fact-checking was performed by editors were crucial factors. Content farms lack credibility and professional editorial systems, and do not actively manage their content. They may post disclaimers to avoid liability. Most of their articles are either plagiarised, copied, or rewritten, and often contain excessive and/or untrue information. However, because their messages are relayed at the same time, this study cross-referenced the data collected from content farms and the electronic versions of the four major news outlets. If the content of an article from a content farm was consistent with its title, it was removed from the content farm message database.

After general and fake news articles were collected, they were classified according to their characteristics. Characteristics indicated by other studies were considered during the

classification process. For example, articles with more adjectives and adverbs, or a writing style that deviates from that of pure news, are usually fake news. Therefore, the system's method for identifying fake news was based on writing style. In addition, the system was developed by referencing the Chinese Knowledge Information Processing (CKIP) Chinese word segmentation system from Academia Sinica in Taiwan (i.e., NLP textual analysis and deep neural network (DNN) learning technology).

### 3.1.2. Design of AI News Credibility Identification System

Python language, TensorFlow tools, machine learning, and deep learning computation were used to test and train the system. By using the NLP tool, multidimensional analysis of word segmentation and participles was performed on the data to develop a system capable of determining news credibility. For data processing, the sources of general and fake news were selected and the required fields for data analysis were designated. Subsequently, information tags required to write web crawler programs were created to collect the required data. Fixed times and frequencies were established, the crawler program was executed daily, and the original data were stored in the database. The system executed the following procedure to analyse the data:

1. Data clean-up: Data were selected from the database to determine whether any of the fields had been left blank. Data with blank fields were excluded. The data were then exported as comma-separated value files containing 8-bit Unicode Transformation Format (UTF-8) order marks and used to train the system in the machine learning process.
2. Feature extraction: the titles and contents of the articles in the source files were formatted as character strings using the CKIP word segmentation system developed by Academia Sinica. The words were segmented and analysed, the textual labels and names were executed, the frequency of each word segmentation was calculated to eliminate anomalies (defined as a frequency of phrases less than 10% of the total count), and the remaining segments were stored as feature parameters.
3. Model building: the system used the Keras sequential model; the sizes and parameters of its input layer, hidden layer, and output layer were fed to the compilation model. The parameters used for the ".model.add" function were as follows: input layer feature dimension (dim) 83; input layer units (8); input layer activation (relu); hidden layer units (8); number of hidden layers (12); hidden layer activation (relu); output layer units and (1); output layer activation (sigmoid).
4. Training and validating of the model: the system used two functions, namely model.compile and model.fit, to train and validate the model. The parameters used for model.compile were: loss function loss (binary_crossentropy); optimised method optimizer (rmsprop); learning curve learning rate (0.001); and training indicator metrics (accuracy). The parameters for model.fit were: batch size batch_size (100); number of epochs to train (100); and validation and split ratio validation_split (0.2).
5. Data visualisation: after model training, the generated data were displayed as a graph using the matplotlib command and stored as an image. The data were placed in the Result data folder.

This study used AI to discriminate among news content and detect invalid news. Language features and message dissemination routes were used as the basis for most of the discrimination. For the language features, a textual analysis of participles, word blocks, word segmentations, and context was performed using NLP. DNN technology was used to develop the system. Unlike machine learning that requires humans to provide rules as learning conditions, DNNs allow systems to conduct self-learning using established databases and parameters in a well-designed neural network.

### 3.1.3. Results and Application of AI News Credibility Identification System

A total of 10,000 data points were collected for the system, comprising 8000 data points used for system training (50% general news and 50% fake news, i.e., 4000 data points

each) and 2000 data points used for testing (50% general news and 50% fake news, i.e., 1000 data points each). The core parameters of the model were dim (83), node (8), lr (0.001), and hidden layer (12). After 100 rounds of training, the highest accuracy according to the experimental results was 90.15% (Table 3).

**Table 3.** The model with the highest accuracy.

| Dim | Node | Lr | Hidden Layer | Epoch | Correct Rate |
|---|---|---|---|---|---|
| 83 | 8 | 0.001 | 12 | 100 | 90.15% |

Figure 1 presents the model's accuracy distribution; the blue line represents accuracy on the training sets and the orange line represents accuracy on the testing sets after 100 rounds of training. The highest accuracy generated in the experiment was 90.15%. Figure 2 presents the model loss distribution, which indicates that the loss stabilised after 100 rounds of training.

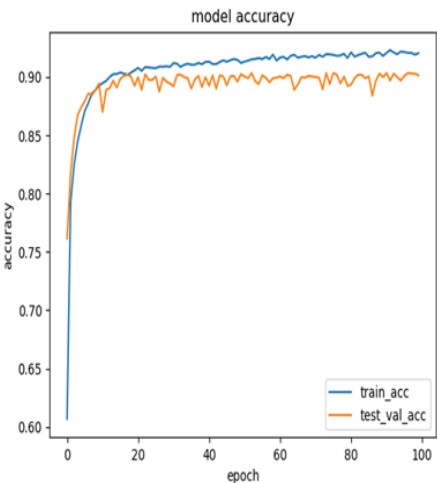

**Figure 1.** Model accuracy.

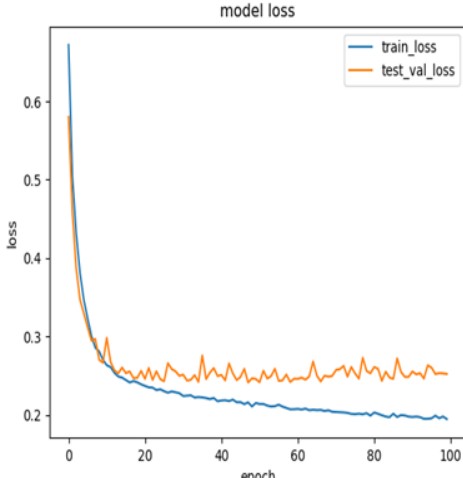

**Figure 2.** Model loss.

Table 4 presents the sources and quantities of the general news messages in the training sets. The sources were selected from the news content database. A total of 5000 random data points from the database were used as the basis for the training system.



**Table 4.** Dataset of training model (normal news).

| Media | Number of Data | Collection Time |
|---|---|---|
| Apple daily | 1250 | Sept. 2019–Mar. 2021 |
| China times | 1250 | Sept. 2019–Mar. 2021 |
| United daily news | 1250 | Sept. 2019–Mar. 2021 |
| Liberty times | 1250 | Sept. 2019–Mar. 2021 |

The training sets contained data from the content farm message database as well. Table 5 presents the quantities and sources of the content farm messages. The use of the content farm messages was complicated in terms of the collection time, as not all of the messages were formatted as news articles and were often written using colloquial words and phrases. In addition, after being banned from platforms or search engines many content farms continue under a different name or domain to avoid regulations. Therefore, the collection of content farm messages was a longer process.

**Table 5.** Dataset of training model (content farms).

| Media | Number of Data | Collection Time |
|---|---|---|
| The Global Times | 1000 | Dec. 2017–Mar. 2021 |
| Mission | 1000 | June 2019–Mar. 2021 |
| Nooho | 1000 | July 2019–Mar. 2021 |
| Kknews | 1000 | Oct. 2019–Mar. 2021 |
| Qiqi.news | 1000 | Nov.2019–Mar. 2021 |

This study prepared 800 news-related data points for testing, comprising 50% general news and 50% fake news. The credibility of each output was identified, and Table 6 presents the results of this experiment. A total of 643 data points were deemed authentic, while 157 were not. The hit rate of the system was 80.375%. At a 95% confidence interval and 2.808% error range, the hit rate range was between 83.183% and 77.567%.

**Table 6.** System test results.

| Test Sample | Hit Rate | 95% Confidence Interval | Error Range |
|---|---|---|---|
| 800 (400 Normal news/400 News from content farms) | 80.375% | 83.183~77.567% | 2.808% |

This study stored the trained model and designed an online user interface. Figure 3 presents an example of the process a user follows to validate content from a CNA news article. Users can input a news article and determine its credibility. The title and contents of the article are required fields, and users must fill in these fields in accordance with the directions next to the fields. The system performs big data computations to generate multiple data points, and features such as textual characteristics, weight, and frequency are determined after word segmentation. These features were selected based on related studies. As the relevant studies describe, one factor that indicates fake news is a higher ratio of adverbs and adjectives. The credibility of a news article appears at the bottom of the interface as reference for the user. Users are reminded that the data should be used as reference only, and that discrimination methods provided by other organisations can be used in addition to the system.

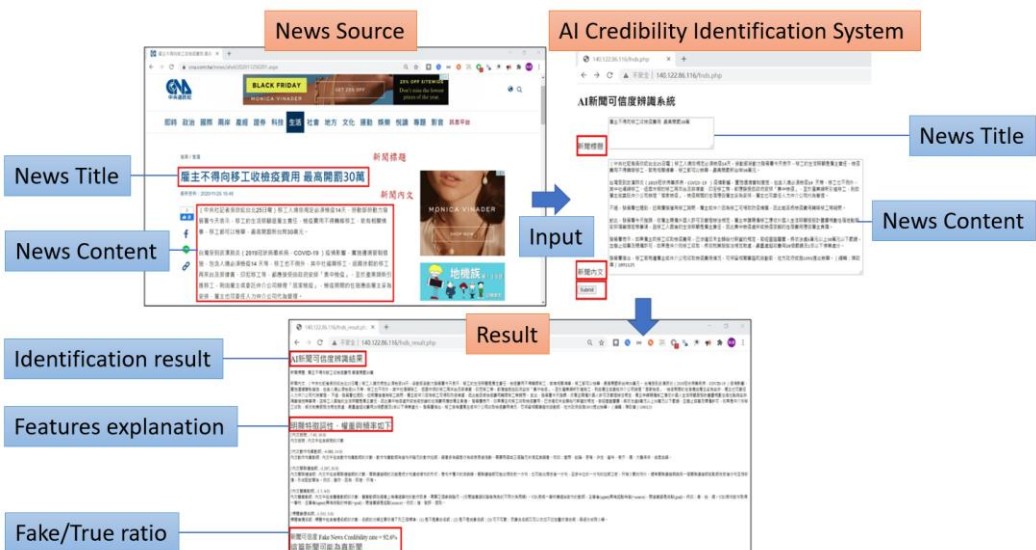

**Figure 3.** Interface of system.

### *3.2. Participants*

This study investigated audiences' attitudes towards news discrimination and the effect of computer-aided evaluation. The research sample comprised 60 undergraduate students in northern Taiwan. The participants were separated into an experimental group (to discriminate news using computer-aided teaching) consisting of 30 subjects and a control group (to discriminate news through other methods) consisting of 30 subjects. Data were used only as a valid sample and reference after the participants' consistency screening.

### *3.3. Research Design*

This study investigated the effectiveness of multimedia computer-aided instruction in solving problems related to media literacy. Concrete suggestions are proposed on the basis of the results of this study, and can serve as a reference for additional studies on fake news and media literacy in the classroom. This study used the news credibility assessment system developed by the Graduate Institute of Mass Communication (hereinafter referred to as 'the Institute') of National Taiwan Normal University as the background, a primary tool of computer-aided instruction, to help the participants discriminate among messages.

#### 3.3.1. Independent Variable

The independent variable was the method of news discrimination. The experimental and control groups received different experimental procedures for discrimination strategies (Figure 4). The procedures were as follows:

1. Experimental group: used the news credibility assessment system developed by the Institute as the basis to discriminate news content. The group was informed of conventional discrimination standards, and the five steps provided by the Harvard Library and the eight steps provided by the IFLA were used as an auxiliary tool for news discrimination.
2. Control group: used the general discrimination method to manually discriminate news content. The five steps provided by the Harvard Library and the eight steps provided by the IFLA were used as the basis for news discrimination.

#### 3.3.2. Dependent Variable

The dependent variable was the news discrimination achievement test. This study developed and administered a news discrimination post-test to the participants, and their scores served as reference for subsequent experiments. The participants were required to complete a questionnaire on learning attitude after the course to identify differences among

their attitudes. The participants who used the system completed another questionnaire regarding their experience with the system. This questionnaire was distributed to collect the participants' suggestions regarding the systems' interface and application, which can serve as a reference for subsequent studies as well as a basis for improving the system.

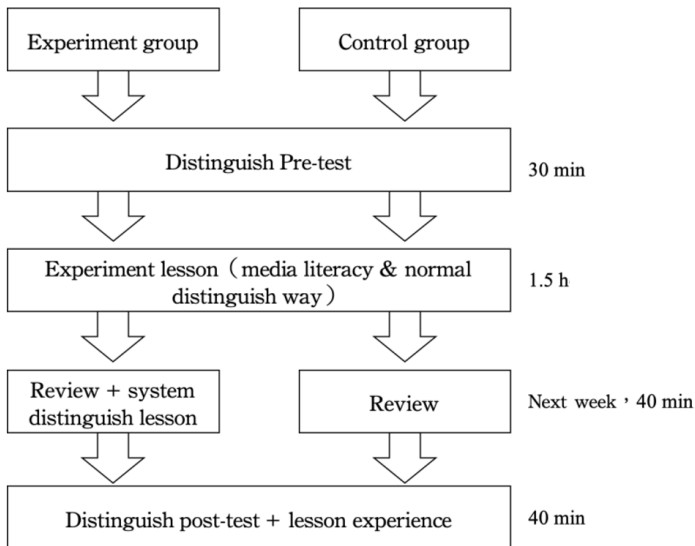

**Figure 4.** Experiment process.

### 3.3.3. Control Variable

The control variables were the instructor, instructional materials for media literacy, duration of instruction and experiment, and assessment instrument. The researcher was the instructor for the two experimental classes. The teaching methods, progress, and tests were the same for both classes. The introduction to the concept of fake news consisted of part of *Digital Literacy for Combating Fake Messages*, written by Professor Hu Yuan-hui and published by the Association for Quality Journalism in 2019, a review of events caused by fake news in Taiwan and abroad, and methods of discriminating fake news provided by international organisations and educational institutions.

### 3.3.4. Dependent Variable

This study conducted a news discrimination test to identify significant differences in the students' scores and their learning experience after they had learned the general discrimination method and the system-based discrimination method. A pre-test and post-test were designed to determine whether the system is able to help students discriminate among news content; both tests consisted of ten current events articles along with headings and a portion of their content for reference. The type and number of questions were the same for both tests. Each question was worth 20 points, and the total possible score was 100 points. The achievement questionnaire had a Cronbach's $\alpha$ of 0.82

Regarding validity, the questions on the achievement test were adapted from the Taiwan FactCheck Center. Fake messages were verified by multiple parties. Messages verified by an impartial third-party organisation were used to avoid conveying the views of a single news agency or personal opinions regarding the agency, which can affect an individual's judgement during the test.

After the experiment, the participants were required to write a reflection paper about the course to determine whether the use of the system in the course affected their attitudes towards learning. The course reflection was revised on the basis of the attitude scale edited by Fennema and Sherman in 1976 [25], with a Cronbach's $\alpha$ of 0.89. The effect of the system on learning attitude was determined by examining the students' perceptions of, attitudes towards, and skills acquired from the media literacy course. A 5-point Likert scale was

used to evaluate attitude; the participants answered each question in accordance with their learning conditions.

### 3.4. Information Processing and Analysis

After the experiment and survey, the questionnaires were verified to identify invalid ones. During this stage, information processing and compilation was performed for the learning attitude scale questions and learning achievement pre-test and post-test; SPSS version 23 was used as an auxiliary tool to determine whether the AI news credibility assessment system affected learning attitude and media literacy.

## 4. Results and Discussion

### 4.1. Analysis of Learning Effectiveness

Thirty students were allocated to the experimental group and thirty to the control group, for a total of sixty participants. For the learning attitude scale (postlesson reflections), sixty valid questionnaires were collected; for the learning achievement tests (pretest and post-test), sixty valid questionnaires were collected as well.

The experimental group scored 38.67, and the control group scored 41.33 (Table 7), each out of a possible score of 100. The pre-test consisted of five questions, each worth twenty points. On the pre-test, the participants in both groups answered only one or two questions correctly; neither group performed exceedingly well. From the perspective of the experiment examples, this result indicated that most of the participants were unfamiliar with message discrimination. Thus, when they received unfamiliar messages, they did not know which message discrimination approach to adopt. A detailed analysis was performed through an independent $t$-test table. The $F$ value of Levene's test was 0.207, $p = 0.651$. Statistical significance was higher than 0.05, indicating that there was no significant difference in pre-test scores observed between the two groups; their performance was similar. We then compared the post-test scores, which are presented in Table 8.

**Table 7.** Grade of Pre-test.

| Type of Test | Group | N | Average | Standard Deviation |
|---|---|---|---|---|
| Pre-test | Experiment group | 30 | 38.67 | 18.114 |
| | Control group | 30 | 41.33 | 22.854 |

**Table 8.** Grade of Post-test.

| Group | N | Average | Standard Deviation |
|---|---|---|---|
| Experiment group | 30 | 86.00 | 14.99 |
| Control group | 30 | 58.00 | 21.24 |

The post-test scores indicated that the experimental group improved more than the control group; their scores increased significantly and almost doubled. The $F$ value of Levene's test was 2.214, $p = 0.142$ (Table 9), lower than 0.05, indicating a significant difference in post-test scores between the two groups. The total possible score was 100, and the test consisted of five questions, each worth twenty points. The experimental group answered one more question correctly than the control group after the experiment. The participants that used the system performed message discrimination more quickly, leaving them more time to double-check the messages. As a result, they answered more questions correctly. The results of the learning achievement test indicate that the experimental group outperformed the control group in terms of message discrimination. Thus, Research Question 1 is supported.

**Table 9.** Post-test independent *t*-test.

| | F | t | df | Sig |
|---|---|---|---|---|
| grade | 2.214 | −5.899 | 58 | 0.000 |
| | | −5.899 | 52.157 | 0.000 |

### 4.2. Analysis of Learning Attitude

The analysis of learning attitude was performed using an attitude scale that the participants completed after the post-test. The scale consisted of thirteen items, with items 10–13 being reverse coded. The scale was used to identify differences in the two groups' answers, and the four aspects of the experimental design were used to examine participants' attitudes towards learning message discrimination. The mean values for the four aspects were scored using a 5-point Likert scale (Table 10). The scores for the reverse-coded items were converted to the corresponding values. After conversion, the mean values of the reverse-coded questions and those in the same aspect were obtained. The questions were based on the methods of the experiment. Table 10 presents the results of the learning attitude assessment.

**Table 10.** Learning attitude.

| | Experiment Group (*n* = 30) | | Control Group (*n* = 30) | |
|---|---|---|---|---|
| | **Mean** | **Sum** | **Mean** | **Sum** |
| Confidence | 3.45 | 103.5 | 3.02 | 90.5 |
| Usefulness | 4.47 | 134.2 | 4.08 | 122.4 |
| Motivation | 4.3 | 129 | 3.87 | 116 |
| Attitude | 4.39 | 131.67 | 3.94 | 118.33 |

The mean values of the experimental group were higher than those of the control group, indicating that experimental attitudes towards learning were affected by the use of the system; thus, the result was positive. The experimental group outperformed the control group, which indicated that the experimental group's use of the system encouraged them to actively discriminate messages.

The independent *t*-test results indicated a significant difference ($p < 0.05$) in the four aspects between the experimental and control groups (Table 11). The groups' answers to learning attitude questionnaire differed because of their different use of the system. Different items were used for each of the four aspects, and attitude towards each aspect differed.

**Table 11.** Learning attitude independent *t*-test.

| | F | t | df | Sig |
|---|---|---|---|---|
| Confidence | 2.737 | −2.398 | 58 | 0.020 |
| Usefulness | 6.927 | −3.394 | 58 | 0.001 |
| Motivation | 0.054 | −2.540 | 58 | 0.014 |
| Attitude | 1.289 | −3.208 | 58 | 0.002 |

For the confidence aspect, as most of the participants had not taken a media literacy course they encountered difficulties in addressing the new topics in the experiment. Both groups scored low on this aspect, indicating the difficulty and complexity of message discrimination. However, the mean scores suggest that the experimental group performed slightly better. Therefore, the system is effective in its use as an auxiliary tool for message discrimination.

For usefulness, the mean of the experimental group was higher than that of the control group, indicating that participants who used the system perceived that message

discrimination is meaningful and warrants attention. These perceptions represent a positive learning outcome.

The motivation aspect included the reverse-coded questions. The experimental group scored higher than the control group, indicating positive learning outcomes. The items for the motivation aspect were related to being careful with news in the future. Because the system serves as a warning system, it causes users to doubt messages. However, this is a key function of the system.

Finally, the attitude measurement reflected whether use of the system led to different results. The system helped the participants discriminate among news sources and caused them to develop a positive attitude towards this difficult task.

The experimental group scored higher than the control group on all of the four aspects of learning attitude. However, no significant difference in confidence was observed between the two groups. This indicates that the participants found news discrimination difficult because most had not previously taken a media literacy course. In addition, the course lasted less than one week. Additional courses should be conducted for longer periods in order to yield more definitive results. The system helped the participants to develop a positive attitude towards news discrimination in terms of the other aspects; the mean values for these aspects support Research Question 2.

The increase in achievement test scores indicates that the participants progressed significantly in message discrimination. The participants who used the system performed better in answering questions about message discrimination. The analysis of the pre-test results revealed that both groups were on a similar level; neither group was more proficient. Hypothesis 2 is related to the participants' attitudes towards learning about message discrimination. The results indicate that the experimental group had a more positive attitude towards learning. Because the two hypotheses were each supported, it can be concluded that the system enabled users to achieve positive outcomes on the achievement and learning attitude tests.

Our results corroborate those of Fishbein and Ajzen (1975), who defined attitude as the degree of influence of an individual's evaluation of a target behaviour, that is, the positive or negative emotions an individual experiences while performing a target behaviour. The more positive an attitude towards message discrimination, the more an individual remembers the concepts conveyed through the experiment and the system.

## 5. Conclusions

Media literacy education provides solutions and strategies for news verification to address the inundation of information from content farms and fake messages. Every country has proposed a clear definition of media literacy to raise public awareness of message discrimination through education and to cultivate the required knowledge and skills. Addressing fake messages can be divided into four strategies: facilitating the establishment of credibility-verifying organisations; supporting new technology for fake message detection; expanding public media; and increasing media literacy. This study determined whether new technologies can strengthen media literacy skills.

Our results can be divided into two parts. The first part consists of the results of the achievement test. After the participants used the system, the mean score of the experimental group was 86 and that of the control group was 58. The experimental group progressed further than the control group (Table 8) and outperformed the control group in answering the test questions. On average, the participants in the experimental group answered one more question correctly than those in the control group, indicating that the system helped them to discriminate among messages. The system offered the experimental group a solid basis for message discrimination as well as the convenience of more time to double-check the messages, as the system provides a concrete value for reference.

The second part of our results consists of participants' attitude towards learning about message discrimination. The experimental group provided more positive feedback in the scales they completed. The mean score of the experimental group for functionality of

message discrimination was 4.47 and that of the control group was 4.08, indicating that the system helped the participants to understand message discrimination (Table 10). The mean of the experimental group in their motivation for message discrimination was 4.30, while that of the control group was 3.86. The items for this aspect were related to warnings for news messages, indicating that the system decreased the participants' aversion to message discrimination and increased their awareness of fake news messages.

The contents of news articles vary widely and may involve professional knowledge and backgrounds. Their credibility often cannot be verified without the required knowledge. However, the system's selection function provides a reference value to compare credibility. This function provides users with more time to reference other information rather than requiring that they understand a topic on the basis of a single article. Users can increase their confidence in message discrimination by practicing the test questions. The mean of the experimental group was 3.45 and that of the control group was 3.02 (Table 10). A significant difference ($p = 0.02$) in these scores was observed. According to the literature, increased confidence positively affects behaviour, indicating that the participants increased their motivation to discriminate among messages after participating in the course and practicing using the system.

The system does have deficiencies. The content displayed by the system is unorganized, and information visibility is poor. This feature must be improved to allow users without a message discrimination background to use the system. In addition, several participants indicated that the items about disseminating knowledge related to media literacy were irrelevant. Because discrimination in news on the basis of parts of speech requires a certain understanding of fake messages and a knowledge of news writing styles, additional courses should enlist teachers with more teaching experience in these areas.

Overall, the results from our two-part experiment indicated that the system positively affected the participants' performance by strengthening their ability to discriminate among messages and deepening their understanding of media literacy. However, the system must continue to evolve; information output should continue to be optimised, database performance should be improved, the interface should be adjusted, and the system should be further incorporated into courses to more effectively facilitate message discrimination and strengthen users' media literacy skills.

**Author Contributions:** Conceptualization, T.H.C.C. and W.-C.W.; methodology, T.H.C.C. and C.-S.L.; software, T.H.C.C. and C.-S.L.; validation, T.H.C.C., W.-C.W. and C.-S.L.; writing, T.H.C.C. All authors have read and agreed to the published version of the manuscript.

**Funding:** This work is supported by the Ministry of Science and Technology, under grants NSC 111-2634-F-003 -002 -, and by the National Taiwan Normal University (NTNU) within the framework of the Higher Education Sprout Project by the Ministry of Education (MOE) in Taiwan.

**Institutional Review Board Statement:** Not applicable.

**Informed Consent Statement:** Written informed consent has been obtained from the patients to publish this paper.

**Data Availability Statement:** Not applicable.

**Conflicts of Interest:** The authors declare no conflict of interest.

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
