# Peer review of "Impact of Artificial Intelligence News Source Credibility Identification System on Effectiveness of Media Literacy Education"

_sustainability, doi:10.3390/su14084830_

Round 1

Reviewer 1 Report

Two brief notes:

1)literature review: while methodology is explained in detail, the review on media literacy is insufficient. The bibliography is scarce and there is a lack of authors such as Sonia Livingstone, Renee Hobbs or Ignacio Aguadeed, among others. It is recommended that the literature review on the subject be reinforced to obtain better grounded conclusions.

2)methodology: The paper could be more robust if we knew how the “attitude” factor was composed. We know that there were 13 questions, but it would be interesting to know which ones to check for homogeneity of the factors and the variance (assuming Varimax rotation was used). This suggestion is not mandatory, but could help the readers.

Author Response

1)literature review: while methodology is explained in detail, the review on media literacy is insufficient. The bibliography is scarce and there is a lack of authors such as Sonia Livingstone, Renee Hobbs or Ignacio Aguadeed, among others. It is recommended that the literature review on the subject be reinforced to obtain better grounded conclusions.

Response: Thank you. Reviewers give authors a very good suggestion. Authors have revised below.

Media literacy has been discussed quite a lot in recent years, but the concept of media literacy was actually put forward in a book by Christ College, Cambridge University, UK in 1950 [1]. At that time, traditional mass media such as radio and television were in a period of vigorous development, and the United Kingdom also began to incorporate media into the school curriculum[2]. Hobbs, R.[3] mainly trains K-12 students to understand "Who produces media contents?", "What is the form of media texts?", "How are media texts produced?", "How do readers understand contents?", "Who are the readers?", "How does the content reproduce the truth?" and so on. In recent years, due to the rapid development of digital technology, media literacy has evolved into digital literacy. Digital literacy education includes: "understanding the content and symbolic characteristics of media messages, speculating on media representation, reflecting on the meaning of readers, analyzing media organization, influence and recent Using Media” et al[4].

  1. Leavis, F. R.; Thompson, D. Culture and environment: The training of critical awareness; Publisher: Chatto & Windus, 1950.
  2. Livingstone, S. Taking risky opportunities in youthful content creation: teenagers' use of social networking sites for intimacy, privacy and self-expression. New Media & Society 2008,10(3), 393-411.
  3. Hobbs, R. Media literacy in action: Questioning the media; Publisher: Rowman & Littlefield, 2021.
  4. Farias-Gaytan, S.; Aguaded, I.; Ramirez-Montoya, M. S. Transformation and digital literacy: Systematic literature mapping. Education and Information Technologies 2022, 27, 1417-1437.

2)methodology: The paper could be more robust if we knew how the “attitude” factor was composed. We know that there were 13 questions, but it would be interesting to know which ones to check for homogeneity of the factors and the variance (assuming Varimax rotation was used). This suggestion is not mandatory, but could help the readers.

Response: Thank you. A good point. Authors will add the next article.

Reviewer 2 Report

The overall quality of the paper is good, only few questions about that:

  1. The reference work needs to be more and with more most recent work, that would enhance the background of the story.
  2. The formats of some figures should be carefully refined, for example, Figure 1 and 2 can be aligned in now row.
  3. Some typos should be checked and corrected.

Author Response

1)The reference work needs to be more and with more most recent work, that would enhance the background of the story.

Response: Thank you. Authors had added some paragraph to literacy review section.

Media literacy has been discussed quite a lot in recent years, but the concept of media literacy was actually put forward in a book by Christ College, Cambridge University, UK in 1950 [1]. At that time, traditional mass media such as radio and television were in a period of vigorous development, and the United Kingdom also began to incorporate media into the school curriculum[2]. Hobbs, R.[3] mainly trains K-12 students to understand "Who produces media contents?", "What is the form of media texts?", "How are media texts produced?", "How do readers understand contents?", "Who are the readers?", "How does the content reproduce the truth?" and so on. In recent years, due to the rapid development of digital technology, media literacy has evolved into digital literacy. Digital literacy education includes: "understanding the content and symbolic characteristics of media messages, speculating on media representation, reflecting on the meaning of readers, analyzing media organization, influence and recent Using Media” et al[4].

Media literacy is the ability to help the public resist the dangers of false information, but it is not easy to cultivate media literacy education. Although European and American countries have begun to attach importance to media literacy education very early, countries such as the United Kingdom, Canada, Australia and other countries have long added media education to it. In the formal system, but in the face of today's fake news problem, the effect is still limited.

  1. Leavis, F. R.; Thompson, D. Culture and environment: The training of critical awareness; Publisher: Chatto & Windus, 1950.
  2. Livingstone, S. Taking risky opportunities in youthful content creation: teenagers' use of social networking sites for intimacy, privacy and self-expression. New Media & Society 2008,10(3), 393-411.
  3. Hobbs, R. Media literacy in action: Questioning the media; Publisher: Rowman & Littlefield, 2021.
  4. Farias-Gaytan, S.; Aguaded, I.; Ramirez-Montoya, M. S. Transformation and digital literacy: Systematic literature mapping. Education and Information Technologies 2022, 27, 1417-1437.

2)The formats of some figures should be carefully refined, for example, Figure 1 and 2 can be aligned in now row.

Response: Thank you. The author has merged the two figures and placed them in the center

3)Some typos should be checked and corrected.

Response: Thank you. Authors had checked again.

Reviewer 3 Report

I certainly believe that the article reaches high levels of quality and honesty in its conclusions, detecting various aspects of improvement in the computer tool designed, which I believe highlights the value of the serious work carried out.
To make a specific comment on the article presented, I would point out that the initial abstract indicates that concern about the risk posed by fake news cannot be associated with events as recent as Brexit or the American presidential elections.
Beyond which I cannot fault the approach made to validate a tool that seems, clearly, more than useful once the improvements suggested in the conclusions are introduced.

Author Response

1)I certainly believe that the article reaches high levels of quality and honesty in its conclusions, detecting various aspects of improvement in the computer tool designed, which I believe highlights the value of the serious work carried out.

Response: Thank you.

2)To make a specific comment on the article presented, I would point out that the initial abstract indicates that concern about the risk posed by fake news cannot be associated with events as recent as Brexit or the American presidential elections.

Response: Thank you. Authors updated the events and had revised below.

Every presidential election, showbiz news, or social events, society began to address the risk of fake news. The Sustainable Development Goals 4 for Global Education Agenda aims to “ensure inclusive and equitable quality education and promote lifelong learning opportunities for all.” by 2030. As a result, various nations have deemed media literacy education a required competence for audiences to maintain a discerning attitude and to verify messages rather than automatically believing messages. This study developed a highly efficient message discrimination method by using new technology.

3)Beyond which I cannot fault the approach made to validate a tool that seems, clearly, more than useful once the improvements suggested in the conclusions are introduced.

Response: Thank you.
